# Extendable and Iterative Structure Learning Strategy for Bayesian Networks

**Hamid Kalantari & Russell Greiner** [*]
Department of Computing Science
University of Alberta
Edmonton, Alberta T6G 2R3, Canada
`{hkalant2,rgreiner}@ualberta.ca`

**Pouria Ramazi**
Department of Mathematics & Statistics
Brock University
St. Catharines, ON L2S 3A1, Canada
`pramazi@brocku.ca`

## Abstract

Learning the structure of Bayesian networks from a dataset of instances is a fundamental yet computationally intensive task, especially as the number of variables grows. Traditional algorithms require retraining from scratch when new variables are introduced, making them impractical for dynamic or large-scale applications. In this paper, we propose an extendable structure learning strategy that efficiently incorporates a new variable $Y$ into an existing (P-map) Bayesian network graph $\mathcal{G}$ over variables $\mathcal{X}$, resulting in an updated P-map graph $\bar{\mathcal{G}}$ on $\bar{\mathcal{X}} = \mathcal{X} \cup \{Y\}$. By leveraging the information encoded in $\mathcal{G}$, our method significantly reduces computational overhead compared to learning $\bar{\mathcal{G}}$ from scratch. Empirical evaluations demonstrate runtime reductions of up to $\mathbf{1300\times}$ without compromising accuracy. Building on this approach, we introduce a novel iterative paradigm for structure learning over $\mathcal{X}$. Starting with a small subset $\mathcal{U} \subset \mathcal{X}$, we iteratively add the remaining variables using our extendable algorithm to construct a P-map graph over the full set. This method achieves runtime advantages compared to common algorithms while maintaining similar accuracy. Our contributions provide a scalable solution for Bayesian network structure learning, enabling efficient model updates in real-time and high-dimensional settings.

## 1 Introduction

Dependency between random variables can be represented by a directed acyclic graph (DAG), where for some directions, a link from a variable A to B signifies that A causes B. When the DAG is coupled with the conditional probability distribution (CPD) of each variable given its parents, it forms a Bayesian network, which enables both probabilistic and causal queries. The joint probability distribution of the variables then factorizes according to the DAG, meaning it becomes the product of the associated CPDs.

Estimating the DAG from observational data, known as structure learning, is typically approached using either constraint-based or score-based algorithms (Kitson et al., 2021). Constraint-based methods, such as PC (developed by Peter Spirtes and Clark Glymour) (Spirtes et al., 2000) and Fast Causal Inference (FCI) (Spirtes et al., 2000), rely on detecting dependencies between variables using conditional independence (CI) tests (Guo et al., 2020). In contrast, score-based methods search for a DAG that maximizes a score function, such as the Bayesian Information Criterion (BIC) (Koller & Friedman, 2009).

A notable gap exists in current approaches: no algorithm efficiently updates an existing DAG when new variables are introduced. This issue is particularly relevant in fields such as social science (Card, 1999), psychology (Primack et al., 2017), and financial studies (Bollen et al., 2011), where important variables may be omitted in the initial stages of research but later recognized as critical. For instance, in stock market prediction models, analysts might begin with historical stock prices, trading volumes, and economic indicators, only to later discover the significant impact of social media sentiment (Bollen et al., 2011). Incorporating such new variables would traditionally require

---

[*]Alberta Machine Intelligence Institute

re-learning the entire DAG, a process that becomes computationally prohibitive as the number of variables grows. While existing online (Kocacoban & Cussens, 2019) and incremental (Alcobe, 2005) structure learning algorithms address scenarios where datasets are updated over time by adding new instances (over the original set of variables), they are not designed to efficiently incorporate a new variable into a learned DAG, as they would, instead, simply discarding the original structure.

We propose an extendable structure learning algorithm that avoids the need to re-learn the entire graph when a new variable is added. Specifically, we investigate the effect of adding a node $Y$ to an already learned *correct* structure $\mathcal{G}$ over a set of variables $\mathcal{X}$. We present two algorithms to obtain the extended structure for $\mathcal{X} \cup \{Y\}$. Our key findings are that adding a new variable results only in the deletion of edges between the original variables, not the addition of new ones, and analyzing how the added node can affect the existing edges between nodes in $\mathcal{X}$.

Consequently, the search for the highest-scoring DAG is confined to a reduced space, rather than the full space of all possible DAGs over the extended set of variables. This reduced search space informs the development of an extendable score-based algorithm, as well as a constraint-based algorithm that leverages the existing CI tests from the learned structure. By significantly reducing the number of CI tests compared to re-learning the structure from scratch, we achieve a more computationally efficient solution. The complexity of the extendable constraint-based algorithm is $\mathcal{O}(NK^m 2^d)$ where $N$ is the cardinality of $\mathcal{X}$, $d$ represents the maximum degree of nodes in $\mathcal{G}$, $m$ is the degree of node $Y$ in true DAG, and $m \leq K \leq N$. This represents a substantial improvement over the PC algorithm's complexity of $\mathcal{O}((N+1)^M)$ where $M = \max\{d, m\}$. Simulation results over some practical benchmark datasets in Bayesian network demonstrate a runtime reduction of up to 1300-fold, while also improving the accuracy of the learned structure in terms of structural Hamming distance.

Furthermore, we also develop an iterative strategy to learn the structure of Bayesian networks using the proposed extendable algorithms. This algorithm initially learns the structure over two randomly-selected random variables, and at each iteration, one of the remaining variables is added to the set using the extendable algorithms. The accuracy and speed of this iterative algorithm are comparable to, and sometimes better than, those of the PC algorithm.

## 2 BACKGROUND

A Bayesian network is a probabilistic graphical model that represents a joint probability distribution over a set of random variables $\mathcal{X} = \{X_1, X_2, \ldots, X_N\}$. In general, the joint distribution $P(\mathcal{X})$ can be factorized using the chain rule as $\Pi_{i=1}^{N} P(X_i \mid X_1, \ldots, X_{i-1})$. Given a network structure, this factorization can be simplified by exploiting conditional independencies among the variables.

Each such factorization corresponds to a Directed Acyclic Graph (DAG) $\mathcal{G}$, where the nodes represent the random variables in $\mathcal{X}$, and edges indicate direct dependencies. Specifically, for each conditional term $P(X_i \mid \mathrm{Pa}_{X_i})$, where $\mathrm{Pa}_{X_i} \subseteq \{X_1, \ldots, X_{i-1}\}$ are the parents of $X_i$, G includes a edge from each parent of $X_i$ to $X_i$ in $\mathcal{G}$. The concept of d-separation in a DAG formalizes the notion of conditional independence among variables. A trail (or path) between two nodes $X$ and $Y$ in $\mathcal{G}$ is a sequence of nodes $(X = X_0, X_1, \ldots, X_n = Y)$ such that each pair $(X_i, X_{i+1})$ is connected by an edge (regardless of direction).The length of a trail is defined as the number of edges it contains, and the distance between two nodes is the minimum length among all trails connecting them. A node $Z$ on a trail is called a *collider* if the edges on the trail meet at $Z$ as $X_{i-1} \rightarrow Z \leftarrow X_{i+1}$.

**Definition 1 (d-separation)** *(Koller & Friedman, 2009) Consider the DAG $\mathcal{G}$ with node set $\mathcal{X}$. A trail $\mathcal{T}$ between two nodes $X$ and $Y$ in $\mathcal{X}$ is* active *relative to a set of nodes $\mathcal{Z}$ if, (i) every non-collider on $\mathcal{T}$ is not a member of $\mathcal{Z}$, and (ii) every collider on $\mathcal{T}$ is an ancestor of some member of $\mathcal{Z}$. The node subsets $\mathcal{X}$ and $\mathcal{Y}$ are* d-separated *given the subset $\mathcal{Z}$, if there is no active trail between any node $X \in \mathcal{X}$ and any node $Y \in \mathcal{Y}$ given $\mathcal{Z}$.*

If $\mathcal{X}$ and $\mathcal{Y}$ are *d-separated* given $\mathcal{Z}$, denoted d-sep$_\mathcal{G}(\mathcal{X}, \mathcal{Y} \mid \mathcal{Z})$, we say that the paths between $\mathcal{X}$ and $\mathcal{Y}$ are *blocked* by $\mathcal{Z}$. Define $\mathcal{I}(\mathcal{G})$ as the set of all d-separations in DAG $\mathcal{G}$. Let $\mathcal{I}(P)$ denote the set of all conditional independencies implied by the distribution $P$. The Markov condition imposes that $\mathcal{I}(\mathcal{G}) \subseteq \mathcal{I}(P)$ and the distribution $P$ is said to be *faithful* to the DAG $\mathcal{G}$ if $\mathcal{I}(P) \subseteq \mathcal{I}(\mathcal{G})$. If $\mathcal{I}(\mathcal{G}) = \mathcal{I}(P)$, as implied by the two assumptions, then $\mathcal{G}$ is called a *P-map (perfect-map)* for $P$. It has been proven that almost all distributions $P$ admit some P-map $\mathcal{G}$ (Koller & Friedman, 2009). A

*P-map learner* is an algorithm, such as PC, that (given a set of instances of the random variables $\mathcal{X}$, drawn from a distribution $P$) attempts to outputs a P-map for that $P$. Should the distribution $P$ not admit a P-map, then the output will be a DAG $\mathcal{G}$, that either violates the faithfulness or Markovness assumption. Two following Lemmas indicate a relation between $\mathcal{I}(\mathcal{G})$ and $\mathcal{I}(P)$.

**Lemma 1** *(Based on (Spirtes et al., 2000)) Consider random variables $\mathcal{X}$ with joint distribution $P$ that admits a P-map $\mathcal{G}$. Vertices $X$ and $Y$ are not adjacent in $\mathcal{G}$ if and only if $X \perp Y \mid \mathcal{U}$ for some (possibly empty) $\mathcal{U} \subseteq \mathcal{X}$.*

**Lemma 2** *[Lemma 3.2 in (Koller & Friedman, 2009)] Consider random variables $\mathcal{X}$ with joint distribution $P$ that admits a P-map $\mathcal{G}$. Vertices $X$ and $Y$ are not adjacent in $\mathcal{G}$ if and only if $X \perp Y \mid \mathrm{Pa}_X$ or $X \perp Y \mid \mathrm{Pa}_Y$.*

## 3 EXTENDABLE LEARNING

Let $\mathcal{X} = \{X_1, \cdots, X_N\}$ be the set of random variables with the joint probability distribution $P'$, and $\mathcal{G}$ be an output of a P-map learner algorithm over $\mathcal{X}$. Now, suppose a new variable $Y$ is added, expanding the variable set to $\bar{\mathcal{X}} = \mathcal{X} \cup \{Y\}$ whose joint distribution is denoted by $P$. We refer to $\bar{\mathcal{X}}$ (resp., $P$) as the *extended* variable set (resp., distribution). Following the common practice in the literature, we assume that there is a P-map for the joint distribution $P$ of the extended variables $\bar{\mathcal{X}}$, but that is not necessarily the case with the joint distribution $P'$ of the original variables $\mathcal{X}$ as explained below. The goal is to efficiently learn a P-map $\bar{\mathcal{G}}$ for $\bar{\mathcal{X}}$, leveraging the information already encoded in $\mathcal{G}$.

**Problem 1** *Consider a set of random variables $\mathcal{X}$ and let $\mathcal{G}$ be the output of a P-map learner applied to $\mathcal{X}$. Consider random variable $Y$ and the extended variable set $\bar{\mathcal{X}} = \mathcal{X} \cup \{Y\}$ with joint distribution $P$. Find a P-map $\bar{\mathcal{G}}$ for $P$.*

A challenge arises because adding $Y$ may alter the dependencies among the variables in $\mathcal{X}$. Specifically, $P'(\mathcal{X})$ is the marginal distribution of $P(\bar{\mathcal{X}})$ over $\mathcal{X}$. However, since $Y$ was unobserved when $\mathcal{G}$ was learned, $\mathcal{G}$ may not accurately capture the dependencies in $P'(\mathcal{X})$. In particular, $\mathcal{G}$ may not be a P-map for $P'(\mathcal{X})$ due to hidden confounding introduced by $Y$. For example, when $Y$ is a confounding variable (hidden common cause) between two collider nodes in $\mathcal{G}$ over $\mathcal{X}$, DAG $\mathcal{G}$ cannot represent all independencies in $P'$, violating the faithfulness assumption (Spirtes, 1995). Consider $\mathcal{X} = \{X_1, X_2, X_3, X_4\}$ and $\bar{\mathcal{G}}$ as $X_1 \rightarrow X_2 \leftarrow Y \rightarrow X_3 \leftarrow X_4$. Marginalization over $Y$ results in two adjacent collider nodes $X_2$ and $X_3$ leading to two immoralities $X_1 \rightarrow X_2 \leftarrow X_3$ and $X_2 \rightarrow X_3 \leftarrow X_4$ in $\mathcal{G}$. Since a valid DAG cannot contain adjacent collider nodes forming multiple immoralities, $\mathcal{G}$ fails to satisfy the P-map condition.

We investigate how adding $Y$ affects the dependencies among the variables in $\mathcal{X}$. Consider two variables $X_1$ and $X_2$ in $\mathcal{X}$. Three possible scenarios can occur when $Y$ is added:

1. **Non-adjacent variables remain non-adjacent:** If $X_1$ and $X_2$ are not adjacent in $\mathcal{G}$, they remain non-adjacent in $\bar{\mathcal{G}}$. by faithfulness, the absence of an edge implies a conditional independence given some subset $\mathcal{U} \subseteq \mathcal{X} \setminus \{X_1, X_2\}$ (Lemma 1), which remains valid even after the inclusion of $Y$.

2. **Spurious adjacencies may be removed:** If $X_1$ and $X_2$ are adjacent in $\mathcal{G}$ but become conditionally independent given $Y$ and some subset $\mathcal{U} \subseteq \mathcal{X} \setminus \{X_1, X_2\}$, the edge between them may be removed in $\bar{\mathcal{G}}$.

3. **True adjacencies remain:** If $X_1$ and $X_2$ are adjacent in $\mathcal{G}$ and remain dependent given $Y$ and any subset $\mathcal{U} \subseteq \mathcal{X} \setminus \{X_1, X_2\}$, the edge between them is preserved in $\bar{\mathcal{G}}$.

According to the first scenario, we show that following Proposition 1 follows from Lemma 1.

**Proposition 1** *Consider $\bar{\mathcal{G}}$ is a P-map over $\bar{\mathcal{X}}$ and $\mathcal{G}$ is a graph over $\mathcal{X}$. If $X_j$ is adjacent to $X_i$ in $\bar{\mathcal{G}}$, then $X_j$ is an adjacent of $X_i$ in $\mathcal{G}$.*

*Proof.* If $X_j$ and $X_i$ are not adjacent in $\mathcal{G}$, Lemma 1 proves there is a subset $\mathcal{U} \subseteq \mathcal{X} \setminus \{X_i, X_j\}$ such that $X_i \perp X_j \mid \mathcal{U}$. Because $\bar{\mathcal{G}}$ is a P-map then the faithfulness assumption means $X_i$ and $X_j$ are d-separated by $\mathcal{U}$ in $\bar{\mathcal{G}}$. Thus there can be no edge between $X_i$ and $X_j$ in $\bar{\mathcal{G}}$. $\qquad\square$

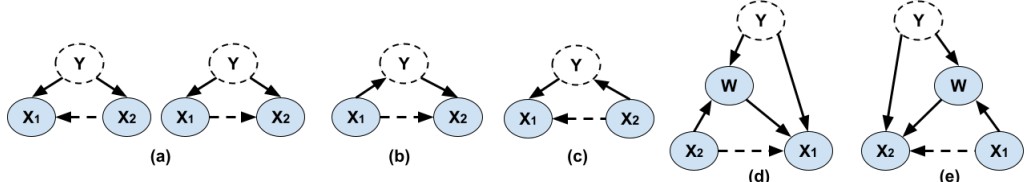

Figure 1: Structures where not observing $Y$ can lead to a spurious edge between $X_1$ and $X_2$. The dashed edge represents the possible direction of the spurious edge when the structure is learned by a P-map learner algorithm and $Y$ is an unobserved node. In (a) either $X_1 \to X_2$ or $X_1 \leftarrow X_2$ can occur, and in the other structures, only one direction can occur.

An important result from Proposition 1 is that adding $Y$ does not introduce new edges between variables in $\mathcal{X}$ that were not already connected in $\mathcal{G}$. Therefore, we only need to examine existing edges in $\mathcal{G}$ and consider potential new edges between $Y$ and the variables in $\mathcal{X}$. We now present our main theoretical results, which characterize how the addition of $Y$ affects the structure of the structure $\mathcal{G}$.

**Lemma 3** *Consider variables $\bar{\mathcal{X}}$ whose joint distribution admits P-map $\bar{\mathcal{G}}$. Let $Y \in \bar{\mathcal{X}}$ and DAG $\mathcal{G}$ be the output of a P-map learner applied to $\bar{\mathcal{X}} \setminus \{Y\}$. Then every pair of non-adjacent nodes $X_1$ and $X_2$ in $\bar{\mathcal{G}}$ are adjacent in $\mathcal{G}$ only if*

1. *$Y$ is a common cause or mediator of $X_1$ and $X_2$ in $\bar{\mathcal{G}}$; or*

2. *$X_1$ is linked to some node $W$ which in turn is linked to $X_2$, and $Y$ is linked to both $W$ and $X_2$ (or the same statement but when $X_1$ and $X_2$ are exchanged).*

*Proof.* $X_1$ and $X_2$ being adjacent in $\mathcal{G}$ implies that they remain dependent conditioned on any subset of $\mathcal{X}$, i.e.,

$$\forall \mathcal{U}' \subseteq \mathcal{X} \qquad X_1 \not\perp X_2 \,|\, \mathcal{U}'. \tag{1}$$

On the other hand, $X_1$ and $X_2$ being non-adjacent in $\bar{\mathcal{G}}$ implies the existence of a subset of $\mathcal{X}$ that together with $Y$ drive $X_1$ and $X_2$ independent, i.e.,

$$\exists \mathcal{U} \subseteq \mathcal{X} \qquad X_1 \perp X_2 \,|\, \mathcal{U} \cup \{Y\}. \tag{2}$$

In view of equation 1, equation 2, and $\bar{\mathcal{G}}$ being a P-map for $\bar{\mathcal{X}}$, it follows that there exists a path $\mathcal{T}$ connecting $X_1$ and $X_2$ in $\bar{\mathcal{G}}$ that is active if $Y$ is not observed, and every path connecting $X_1$ and $X_2$ becomes inactive if $Y$ and $\mathcal{U}$ are observed. The distance of $Y$ to $X_1$ does not exceed two. Otherwise, for every path $\mathcal{T}_i$ connecting $Y$ to $X_1$, let $W_i$ be the neighbor of $X_1$ on $\mathcal{T}_i$ and $V_i$ be the neighbor of $W_i$ on $\mathcal{T}_i$. If $Y$ is not a parent for $X_2$, considering Lemma 2, $Y$ cannot impact the existence of an edge between $X_1$ and $X_2$. If $Y$ is a parent of $X_2$, then two cases must be checked. (*1*) There is no collider node $V_j$ between $X_1$ and $X_2$ by observing $W_i$ or $V_i$ the path will be inactive and the other paths are blocked by parent nodes of $X_1$ or $X_2$. (*2*) There is a collider node $V_j$ between $X_1$ and $X_2$, with children $W_i$ and $V_i$. In this case, either $V_i$ is a collider on the path between $Y$ and $X_1$, or there is a collider between $V_i$ and $Y$ that blocks the path; otherwise, a cycle would be formed in the graph. Therefore, two conditions are possible. (*i*) $Y$ must be an adjacent of both $X_1$ and $X_2$ such that $Y$ is not a collider between them, or (*ii*) $Y$ must be adjacent of one of them and has a trail with length two to the other. (*i*) is the same as case (*1*). For (*ii*), if there are other structures except what is indicated in Fig.1(d,e) would be led to $X_1 \perp X_2 \,|\, \mathcal{U}'$ for a $\mathcal{U}' \subset \mathcal{X}$, which is contrary to (1). $\qquad \square$

Considering Lemma 3, only three cases exist where observing $Y$ in $\bar{\mathcal{G}}$ can remove the edge between $X_1$ and $X_2$: When $Y$ is a confounding (Fig. 1 (a)) or mediator variable between them (Fig. 1 (b,c)) or $Y$ is adjacent to $X_1$ and forms a collider with $X_2$ and $W$ while $W$ is a mediator node between $X_1$ and $X_2$ (Fig. 1 (d,e)). Only in these cases is there an active path between $X_1$ and $X_2$ when $Y$ is not observed and where that path is blocked by observing $Y$ in $\bar{\mathcal{G}}$.

**Lemma 4** *Let $\bar{\mathcal{G}}$ be a P-map for $P$. If $X_1 \perp X_2 | \mathcal{U}$ for $\mathcal{U} \subset \mathcal{X} \setminus \{X_1, X_2\}$, then $Y$ cannot be a mediator or common cause variable between them in $\bar{\mathcal{G}}$.*

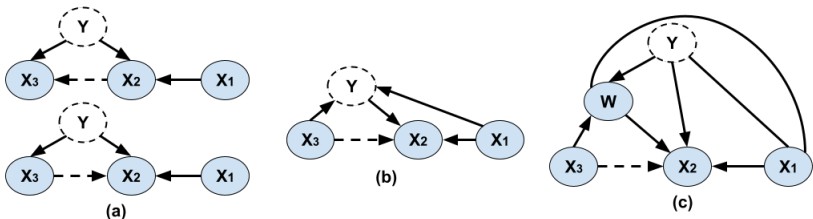

Figure 2: Three structures that show the effect of the unobserved node $Y$ on an immorality

*Proof.* Consider $Y$ as a mediator or common cause variable between $X_1$ and $X_2$. Then the path $X_1 \rightleftharpoons Y \rightleftharpoons X_2$ is active when $Y$ is a hidden variable and $X_1 \not\perp X_2 | \mathcal{U}$ for all $\mathcal{U} \subset \mathcal{X} \setminus \{X_1, X_2\}$. $\square$

**Lemma 5** *Let $\bar{\mathcal{G}}$ be a P-map of $P$ over $\bar{\mathcal{X}} = \mathcal{X} \cup \{Y\}$ and $\mathcal{G}$ is the output of a P-map learner algorithm. If $X \in \mathcal{X}$ is a collider node in $\mathcal{G}$, then it is a collider node in $\bar{\mathcal{G}}$.*

*Proof.* Consider an immorality $X_1 \rightarrow X_2 \leftarrow X_3$ in $\mathcal{G}$. There is a $\mathcal{U} \subset \mathcal{X} \setminus \{X_1, X_3\}$ so that $X_2 \notin \mathcal{U}$ and $X_1 \perp X_3 | \mathcal{U}$. Also, for all $\mathcal{U} \subseteq \mathcal{X} \setminus \{X_1, X_3\}$ we have $X_1 \not\perp X_3 | X_2, \mathcal{U}$. Then $X_1 \not\perp X_3 | X_2, \mathcal{U}, Y$. If $X_1, X_2$ and $X_2, X_3$ are adjacent in $\bar{\mathcal{G}}$, then they form an immorality in $\bar{\mathcal{G}}$ and $X_1 \rightarrow X_2 \leftarrow X_3$ appears in $\bar{\mathcal{G}}$. Now, consider the edge $X_2 \leftarrow X_3$ is removed by observing $Y$. Therefore, $X_2, X_3$, and $Y$ may form one of the structures shown in Fig. 1. Of course, because $X_1 \not\perp X_3 | X_2$, we cannot have a direct path as $X_2 \rightarrow Y \rightarrow X_3$. As a result, three types of structures might occur. If $Y$ is a confounding variable for $X_2$ and $X_3$ (Fig. 2 (a)) Lemma 4 means there is no edge between $Y$ and $X_1$ and we have $X_1 \not\perp X_3 | X_2$ which in turn means $X_2$ must be a collider node between $Y$ and $X_1$ and the edge between $X_1, X_2$ orients as $X_1 \rightarrow X_2$ in $\bar{\mathcal{G}}$. In the second case, if we have $X_3 \rightarrow Y \rightarrow X_2$ (Fig. 2 (b)), similar to the previous case, we have $X_1 \rightarrow X_2$. Also, if $X_1$ and $Y$ are adjacent, the edge between them orients as $X_1 \rightarrow Y$ due to Lemma 4. In third case (Fig. 2 (c)), if $X_1 \leftarrow X_2$ then we have a direct path $X_3 \rightarrow W \rightarrow X_2 \rightarrow X_1$ that can be blocked by observing $X_2$ or $X_3 \perp X_1 | X_2$ which is a contradiction and similar to two first cases we have $X_1 \rightarrow X_2$. Since we have $X_3 \not\perp X_1 | X_2, \mathcal{U}$ it is impossible to have a structure with $X_3$ adjacent to $Y$ and $W$ a mediator node between $X_2$ and $X_3$ as $X_2 \rightarrow W \rightarrow X_3$. If we have an immorality $X_1 \rightarrow X_2 \leftarrow X_3$ but the edges between $X_1, X_2$ and $X_3, X_2$ are removed by observing $Y$, so there are direct paths $X_3 \rightarrow Y \rightarrow X_2$ or $X_3 \rightarrow W \rightarrow X_2$, and $X_1 \rightarrow Y \rightarrow X_2$ or $X_1 \rightarrow V \rightarrow X_2$ in $\bar{\mathcal{G}}$ for some $W, V \in \mathcal{X}$. Therefore, the direction between $X_1, X_2$ or $X_3, X_2$ does not change when $Y$ is added to variables. As a result, the orientations of immoralities in $\mathcal{G}$ will be unchanged in $\bar{\mathcal{G}}$, and so all orientations in $\bar{\mathcal{G}}$ between nodes in $\mathcal{X}$ are similar to the orientations in $\mathcal{G}$. $\square$

### 3.1 CONSTRAINT-BASED APPROACH

Checking CI tests to detect independencies is the main idea in constraint-based algorithms. Two steps are required to add the new variable $Y$ to the previous structure. The first step is checking the relation between $Y$ and other nodes in $\mathcal{X}$, and the second step is investigating the effect of $Y$ on the edges in the previous structure.

The PC algorithm is one of the most popular constraint-based algorithms to learn such structures. Algorithm 1 is an extendable version of the PC algorithm. According to the PC algorithm, the quantity of CI tests required to verify the existence of an edge between two nodes is directly proportional to the number of adjacent nodes in the true DAG. Hence, to identify the existence of an edge between $Y$ and $X \in \mathcal{X}$, it is necessary to perform CI tests between $Y$ and $X$ while conditioning on all subsets of both adjacents of $X$ and $Y$ in $\hat{\mathcal{G}}$, which are indicated by $\mathrm{Adj}(\hat{\mathcal{G}}, X)$ and $\mathrm{Adj}(\bar{G}, Y)$, respectively. Moreover, based on the PC algorithm, by adding the node of $Y$ into the graph $\hat{\mathcal{G}}$, all nodes in $\mathcal{X}$ must be connected to $Y$, forming an initial graph $\bar{\mathcal{G}}$. Subsequently, the process involves refining the graph by eliminating any surplus or spurious edges. The count of adjacent nodes to $Y$ is $| \mathrm{Adj}(\bar{G}, Y) | = N$, whereas $| \mathrm{Adj}(\bar{G}, X) | \leq N$ for any $X \in \mathcal{X}$. Consequently, to determine the existence of edges between $Y$ and each $X \in \mathcal{X}$, it is appropriate to initially conduct CI tests on $\mathcal{U} \subset \mathrm{Adj}(\bar{G}, X)$ and subsequently on $\mathcal{U} \subset \mathrm{Adj}(\bar{G}, Y)$. Once the true edges between $Y$ and all $X \in \mathcal{X}$ are detected, we can then identify the spurious edges between $X, Z \in \mathcal{X}$.

If $d$ represents the maximum degree of nodes in $\hat{\mathcal{G}}$, and $m$ is the degree of node $Y$ in true DAG, employing the PC algorithm for all nodes in $\bar{\mathcal{X}} = \mathcal{X} \cup \{Y\}$ imposes a bound on the number of CI tests, which is $(N+1)^{M+1}$ where $M = \max\{d, m\}$. This bound is established because the PC algorithm does not leverage information from the prior graph. However, applying the Extendable PC algorithm when adding a new variable to the variable set can mitigate the number of required CI tests. Table 1 illustrates the count of CI tests at each step in the Extendable PC algorithm. $N2^d$ (resp., $md2^d$) constrain the number of CI tests in steps 2 (resp. 4), and step 3 may require up to $K^m$ CI tests, where $K \in \{m, m+1, \cdots, N\}$ denotes the number of adjacents of $Y$ after step 2. Nevertheless, in step 2, certain edges between $Y$ and other nodes may be eliminated. If the number of nodes adjacent to $Y$ decreases, the number of conditional independence tests will accordingly decrease in step 3. As a result, we have proved that the number of CI tests for the Extendable PC algorithm is always fewer than the PC one; see Proposition 2. In addition, Theorem 1 proves the output of Algorithm 1 is a P-map.

---

**Algorithm 1:** The Extendable PC Algorithm

---

**Input:** A new variable $Y$ and graph $\hat{\mathcal{G}}$ obtained from the PC algorithm over $\mathcal{X}$;
**Output:** New graph $\bar{\mathcal{G}}$ over the set of variables $\bar{\mathcal{X}} = \mathcal{X} \cup \{Y\}$;

1 Connect $Y$ to all nodes in $\hat{\mathcal{G}}$ and construct the graph $\bar{\mathcal{G}}$;
2 $\text{Adj}(\bar{\mathcal{G}}, Y) = \mathcal{X}$;         // Step 1:  Initializing $\bar{\mathcal{G}}$ and the adjacent sets
3 $\text{Adj}(\bar{\mathcal{G}}, X) = \text{Adj}(\hat{\mathcal{G}}, X) \cup \{Y\}$, for all $X \in \mathcal{X}$; $\text{Sepset}(X, Y) = \emptyset$ for $X \in \mathcal{X}$;
4 $m = 0$
5 **while** *maximum degree of nodes $\mathcal{X}$ in $\bar{\mathcal{G}}$ is greater than $m$* **do**
    // Step 2:  Checking edges between the new variable and other nodes by conditioning on the neighbors of nodes in $\mathcal{X}$.
6    **for** $X \in \mathcal{X}$
7        **for** $U \subseteq \text{Adj}(\hat{\mathcal{G}}, X)$ *and* $| U | = m$
8            **if** $X \perp Y \mid U$
9                Remove the edge $X - Y$ from $\bar{\mathcal{G}}$;
10               $\text{Sepset}(X, Y) \leftarrow U$;
11   $m = m + 1$;
12 $m = 0$;
13 **while** *degree of $Y$ in $\bar{\mathcal{G}}$ is greater than $m$* **do**
    // Step 3:  Checking the edges between the new node and its neighbors by conditioning on the neighbors of the new node.
14   **for** $X \in \text{Adj}(\bar{\mathcal{G}}, Y)$
15       **for** $U \subseteq \text{Adj}(\bar{\mathcal{G}}, Y) \setminus \{X\}$ *and* $| U | = m$
16           **if** $Y \perp X \mid U$
17               Remove the edge $X - Y$ from $\bar{\mathcal{G}}$;
18               $\text{Sepset}(X, Y) \leftarrow U$;
19   $m = m + 1$;
20 $m = 0$;
21 **while** *maximum node degree in $\bar{\mathcal{G}}$ is greater than $m$* **do**
    // Step 4:  Checking edges between nodes in $\mathcal{X}$ with observing new variable $Y$
22   **for** $X \in \text{Adj}(\bar{\mathcal{G}}, Y)$
23       **for** $Z \in \text{Adj}(\bar{\mathcal{G}}, X) \setminus \{Y\}$
24           **if** $Z \in \text{Adj}(\bar{\mathcal{G}}, Y)$ *or* $\text{Adj}(\bar{\mathcal{G}}, X) \cap \text{Adj}(\bar{\mathcal{G}}, Z) \cap \text{Adj}(\bar{\mathcal{G}}, Y) \neq \emptyset$
25               **for** $U \subseteq \text{Adj}(\bar{\mathcal{G}}, X) \setminus \{Z\}$ *and* $| U | = m$
26                   **if** $X \perp Z \mid \{Y\} \cup U$
27                       Remove the edge $X - Z$ from $\bar{\mathcal{G}}$;
28                       $\text{Sepset}(X, Z) \leftarrow U$;
29   $m = m + 1$;
30 **if** $X, Z \in \text{Adj}(\bar{\mathcal{G}}, Y)$, *and* $X \notin \text{Adj}(\bar{\mathcal{G}}, Z)$   // Step 5 :  Immorality detection
31   **if** $X \not\perp Z \mid Y$ *and* $Y \notin \text{Sepset}(X, Z)$
32       Orient $X \rightleftharpoons Y \rightleftharpoons Z$ as $X \rightarrow Y \leftarrow Z$.
33 Orient the other edges by orientation rules in (Spirtes et al., 2000).           // Step 6

Table 1: The number of CI tests for each step of the Extendable PC Algorithm

| Step | 2 | 3 | 4 |
|---|---|---|---|
| Number of CI tests | $\mathcal{O}(N2^d)$ | $\mathcal{O}(K^m)$ | $\mathcal{O}(md2^d)$ |

**Proposition 2** *The number of CI tests of Algorithm 1 is fewer than the PC algorithm.*

**Theorem 1** *The output of Algorithms 1 and 4 is a P-map.*

*Proof.* The proof is a straightforward conclusion using Lemma 1, Lemma 3, and Lemma 4 (in Appendix). Lemma 4 shows that adding a new variable cannot add an edge between two nodes. Hence, according to Lemma 1, the output skeleton of the proposed extendable algorithm finds the skeleton of the true DAG. Then, Lemma 3 shows that all collider nodes were found correctly by the proposed algorithm. So the output PDAG for constraint-based algorithms, such as Algorithm 1, is a P-map structure. □

In addition, we use a straightforward modification of the PC algorithm using Proposition 1. As discussed above, adding a new variable cannot add any edge to the previous structure. Therefore, we can use the previous skeleton $\mathcal{G}$ as the input graph of the PC algorithm and check the other CI tests to obtain $\bar{\mathcal{G}}$. This algorithm is called the *Initialized PC* algorithm (IPC).

### 3.2 SCORE-BASED APPROACH

In the score-based approach, a score function is used to find an optimal structure over all possible DAGs, or perhaps a sub-optimal solution over a subset of possible DAGs. Therefore, the number of DAGs in the search space has a key role in the complexity of the structure learning algorithm. If a DAG $\hat{\mathcal{G}}$ was obtained by a score-based algorithm over $\mathcal{X}$, the search space for learning a new structure that includes $Y$ could be estimated by Proposition 1 and Lemmas 3-5. This means the number of DAGs in this search space will be lower than all possible DAGs on $\bar{\mathcal{X}}$. Let $\mathcal{S}_{\bar{\mathcal{X}}}$ be a search space on $\bar{\mathcal{X}}$. The DAGs $\bar{\mathcal{G}}$ in $\mathcal{S}_{\bar{\mathcal{X}}}$ must satisfy following conditions:

1. If $X_i, X_j \in \mathcal{X}$ are not adjacent in $\mathcal{G}$, then they are not in $\bar{\mathcal{G}}$.

2. If $X_i \in \mathcal{X}$ is a collider node in $\mathcal{G}$, then it is a collider node in $\bar{\mathcal{G}}$.

3. If $X_i, X_j$ are adjacent to each other in $\mathcal{G}$, if $X_i, X_j$ and $Y$ form a structure similar to one of the structures in Fig. 1, then the edge between them can be deleted in $\bar{\mathcal{G}}$.

4. If $X_i, X_j \in \mathcal{X}$ are not adjacent to each other in $\mathcal{G}$, and both of them are adjacent to $Y$ in $\bar{\mathcal{G}}$, then $Y$ must be a collider (i.e., $X_i \rightarrow Y \leftarrow X_j$ in $\bar{\mathcal{G}}$).

5. If $X_i, X_j \in \mathcal{X}$ are adjacent to each other in $\mathcal{G}$, and both of them are collider nodes in $\mathcal{G}$, then $Y$ must be a confounding variable as $X_i \leftarrow Y \rightarrow X_j$ in $\bar{\mathcal{G}}$.

Algorithms 2 and 8 are developed for extendable score-based structure learning approach. Algorithm 2 represents a general extendable score-based algorithm that includes:(1) a search space trimming function (T-function in Algorithm 8) that restricts the graph search space, based on the analysis from Lemmas 3 - 5; and (2) a score-based P-map learner (for example global minimization of the BIC score), that finds the best graph within the restricted search space.

---

**Algorithm 2:** The Extendable Score-based Algorithm

**Input:** A new variable $Y$ and a structure $\hat{\mathcal{G}}$ over $\mathcal{X}$
**Output:** A P-map $\bar{\mathcal{G}}$ over $\bar{\mathcal{X}} = \mathcal{X} \cup \{Y\}$

```
1 S_X̄ ← T(Ĝ, Y, S_X̄)              // By T-function in algorithm 8
2 Ḡ ← PF(S_X̄)                     // PF is a score-based P-map learner
```

---

### 3.3 ITERATIVE STRUCTURE LEARNING APPROACH

We developed a new structure learning paradigm using the extendable approach, allowing standard algorithms to be modified to reduce the run-time. This is achieved through an iterative process where the extendable structure learning algorithm is applied at each step. As shown in Algorithm 3, starting with two randomly selected variables from $\mathcal{X}$, denoted as $X_1$ and $X_2$, a structure $\mathcal{G}_1$ is learned. Then, a third variable $X_3$ is selected from $\mathcal{X} \setminus \{X_1, X_2\}$, and a new structure $\mathcal{G}_2$ is formed by incorporating $X_3$ using the extendable algorithm. This process is repeated iteratively, with each new variable, such as $X_4 \in \mathcal{X} \setminus \{X_1, X_2, X_3\}$, being added to the current set to form the next structure. The procedure continues until all $N$ variables are included, resulting in a P-map graph over $\mathcal{X}$. Using an iterative approach, at each step, we leverage information about the relationships between nodes from the previous graph to determine the current graph. Since the number of nodes impacts the number of CI tests, fewer nodes result in fewer CI tests when applying Lemma 1, which restricts the space of possible graphs. Also, the performance of the iterative algorithms depends on the order in which variables are selected. According to Lemma 1 and Figure 1, if the ordering is close to topological causal ordering the performance of the iterative will be closer to the standard algorithm. For example, consider a naive Bayes structure with $n$ children $\{X_1, \cdots, X_n\}$ and a parent node $Y$. With ordering like $\langle X_1, \cdots, X_n, Y \rangle$, before dealing with $Y$, the iterative algorithm will first produce the complete graph over $\{X_1, \cdots, X_n\}$. However, with this ordering $\langle Y, X_1, \cdots, X_n \rangle$ the number of CI tests for Algorithm 3 will be fewer than the previous ordering.

---

**Algorithm 3:** The Iterative P-map learner Algorithm

---

**Input:** A set of variables $\mathcal{X}$ and their joint probability distribution $P$
**Output:** A partially directed acyclic graph

1  $\hat{\mathcal{X}} = \{X_1, X_2\}$
2  $\mathcal{G} \leftarrow$ P-map learner($\hat{\mathcal{X}}$)
3  **while** $\mathcal{X} \setminus \hat{\mathcal{X}} \neq \emptyset$ **do**
4      Select $X \in \mathcal{X} \setminus \hat{\mathcal{X}}$
5      $\bar{\mathcal{G}} \leftarrow$ Extendable P-map learner($\mathcal{G}, X$)
6      $\hat{\mathcal{X}} \leftarrow \hat{\mathcal{X}} \cup \{X\}$
7      $\mathcal{G} \leftarrow \bar{\mathcal{G}}$

---

## 4 NUMERICAL RESULTS

We now compare the results of our Extendable PC algorithm with the PC, and Initialized PC algorithms on the data sets ASIA (Lauritzen & Spiegelhalter, 1988), CANCER (Korb & Nicholson, 2010), SURVEY (Scutari & Denis, 2021), EARTHQUAKE (Korb & Nicholson, 2010), ALARM (Beinlich et al., 1989), INSURANCE (Binder et al., 1997), CHILD (Spiegelhalter & Cowell, 1992), WATER (Jensen et al., 1989), SACHS (Jensen & Jensen, 2013), MILDEW (Jensen & Jensen, 2013), WIN95PTS (Jensen & Jensen, 2013), HEPAR2 (Onisko, 2003), and ANDES (Conati et al., 1997). For each task, we first draw 10000 instances from distributions for use in structure learning algorithms. For each data set a variable is chosen randomly and a structure is learned over the other variables by the PC algorithm. Then the chosen variable is added to the data set, and the learned structure is considered as the input of the Extendable PC algorithm and Initialized PC to learn the new structure. For iterative PC, the first two variables are chosen randomly, and the iterative PC is used to estimate the structure over the whole of variables. The number of CI tests for the PC, Initialized PC, and Extendable PC algorithms are shown in Table 2 and the runtime in Table 3. In addition, by considering the structural Hamming distance, we recorded the number of incorrect edges either missing or extra compared to the true graph and divided it by the total number of edges in the true DAG (Table 4). These results suggest that the extendable approach can significantly reduce both the number of required CI tests and the runtime, particularly in large networks. Additionally, Table 5 shows the number of CI tests for iterative PC and PC algorithms, and Tables 6 and 7 illustrate the runtime and error of them. The iterative approach applied to the PC algorithm demonstrates the runtime across most datasets is less than the standard PC algorithm, and the error did not change.

Table 2: Number of CI tests

| DATASET | EXTENDABLE PC | INITIALIZED PC | PC |
|---|---|---|---|
| EARTHQUAKE | **15** | 48 | 57 |
| CANCER | **12** | 39 | 45 |
| SURVEY | **11** | 49 | 55 |
| ASIA | **26** | 87 | 124 |
| CHILD | **184** | 1242 | 2124 |
| SACHS | **618** | 682 | 971 |
| ALARM | **103** | 745 | 3283 |
| MILDEW | **200** | 670 | 3629 |
| WIN95PTS | **86** | 1975 | 12501 |
| INSURANCE | **147** | 1571 | 5078 |
| WATER | **71** | 278 | 1346 |
| HEPAR2 | **536** | 5108 | 23202 |
| ANDES | **277** | 11426 | 68375 |

Table 3: Run-Time (sec)

| DATASET | EXTENDABLE PC | INITIALIZED PC | PC |
|---|---|---|---|
| EARTHQUAKE | **0.033** | 0.116 | 0.283 |
| CANCER | **0.029** | 0.115 | 0.294 |
| SURVEY | **0.022** | 0.186 | 0.425 |
| ASIA | **0.071** | 0.244 | 0.744 |
| CHILD | **6.11** | 33.76 | 65.71 |
| SACHS | **15.39** | 15.86 | 34.16 |
| ALARM | **1.45** | 17.95 | 22.10 |
| MILDEW | **28.13** | 31.01 | 316 |
| WIN95PTS | **0.688** | 77.81 | 111 |
| INSURANCE | **2.36** | 36.9 | 58.28 |
| WATER | **0.289** | 1.38 | 5.77 |
| HEPAR2 | **47.57** | 474 | 1832 |
| ANDES | **1.97** | 532 | 2652 |

Table 4: Structural Hamming Distance divided by the total number of true edges (%)

| DATASET | EXTENDABLE PC | INITIALIZED PC | PC |
|---|---|---|---|
| EARTHQUAKE | 0 | 0 | 0 |
| CANCER | 0 | 0 | 0 |
| SURVEY | 0 | 0 | 0 |
| ASIA | 12.5 | 12.5 | 12.5 |
| CHILD | 4 | 4 | 4 |
| SACHS | 0 | 0 | 0 |
| ALARM | 8.7 | 8.7 | 8.7 |
| MILDEW | **13** | 17.4 | 17.4 |
| WIN95PTS | 38.4 | 38.4 | 38.4 |
| INSURANCE | 30.8 | 30.8 | 30.8 |
| WATER | **57.6** | 59.1 | 59.1 |
| HEPAR2 | 51.2 | 51.2 | 51.2 |
| ANDES | 19.5 | 19.5 | 19.5 |

## 5 CONCLUSION

The proposed extendable structure learning approach allows us to add new variables to the given (correct) Bayesian network, with a significantly lower computational burden compared with learning a new structure from scratch. The proposed approach can be applied to all constraint-based and score-based algorithms. A key challenge is using P-map learner algorithms in the presence of a hidden variable. In this case, the output of the algorithms is not a P-map and even in some situations the

faithfulness assumption is violated. We proposed Lemmas to detect situations in which unfaithfulness can occur while there is an unobserved variable. Then, we proposed an extendable strategy for constructing a P-map when a new variable is added to the set of variables. We applied the extendable approach to the PC algorithm, then shows that this algorithm could reduce the runtime up to 1300 times compared with the PC when a new variable is added to the set of variables and up to 270 times compared with the Initialized PC algorithm. In addition, the iterative paradigm for structure learning based on the extendable approach, and demonstrated that this approach can be used effectively for all types of structure learning algorithms. The structure learning starts with two variables and then incrementally adds one variable at a time, to the previous structure using the extendable approach, until all variables are added to the graph. The iterative PC algorithm can reduce the number of CI tests and the runtime for most datasets, while also increasing accuracy in some cases. For future work, optimizing variable ordering to reduce the number of CI tests in the iterative approach could enhance efficiency.

Table 5: Number of CI tests

| DATASET | ITERATIVE PC | PC |
|---|---|---|
| EARTHQUAKE | **31** | 57 |
| CANCER | **27** | 45 |
| SURVEY | **37** | 55 |
| ASIA | **66** | 124 |
| CHILD | 3344 | **2124** |
| SACHS | 1276 | **971** |
| ALARM | 4847 | **3283** |
| MILDEW | **2597** | 3629 |
| WIN95PTS | **10412** | 12501 |
| INSURANCE | **2589** | 5078 |
| WATER | **872** | 1346 |
| HEPAR2 | **8371** | 23202 |
| ANDES | **35327** | 68375 |

Table 6: Run-Time (sec)

| DATASET | ITERATIVE PC | PC |
|---|---|---|
| EARTHQUAKE | **0.09** | 0.283 |
| CANCER | **0.06** | 0.294 |
| SURVEY | **0.09** | 0.425 |
| ASIA | **0.15** | 0.744 |
| CHILD | 124 | **65.71** |
| SACHS | 35.5 | **34.16** |
| ALARM | 81 | **22.1** |
| MILDEW | 715 | **316** |
| WIN95PTS | 144 | **111** |
| INSURANCE | **39.5** | 58.28 |
| WATER | **3.11** | 5.77 |
| HEPAR2 | **597** | 1832 |
| ANDES | **307** | 2652 |

Table 7: Structural Hamming Distance divided by the total number of true edges (%)

| DATASET | ITERATIVE PC | PC |
|---|---|---|
| EARTHQUAKE | 0 | 0 |
| CANCER | 0 | 0 |
| SURVEY | 0 | 0 |
| ASIA | 12.5 | 12.5 |
| CHILD | 4 | 4 |
| SACHS | 0 | 0 |
| ALARM | 17.4 | **8.7** |
| MILDEW | 54.3 | **17.4** |
| WIN95PTS | **31.25** | 38.4 |
| INSURANCE | **26.9** | 30.8 |
| WATER | **47** | 59.1 |
| HEPAR2 | **46.3** | 51.2 |
| ANDES | 23.4 | **19.5** |

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

## A  APPENDIX

---

**Algorithm 4:** The Extendable Constraint-based Algorithm

---

**Input:** A new variable $Y$ and a structure $\hat{\mathcal{G}}$ over $\mathcal{X}$
**Output:** A PDAG $\bar{\mathcal{G}}$ over $\bar{\mathcal{X}} = \mathcal{X} \cup \{Y\}$

1 Form the $\bar{\mathcal{G}}$ over nodes $\bar{\mathcal{X}}$ by connecting $Y$ to all nodes $\hat{\mathcal{G}}$ by undirected edge;
2 **for** $X \in \mathcal{X}$          `// Step 1:`
3    Check the edge between $Y$ and $X$
4 **for** $X \in \text{Adj}(\bar{\mathcal{G}}, Y)$          `// Step 2`
5    **for** $Z \in \text{Adj}(\bar{\mathcal{G}}, X)$
6      **if** $Z \in \text{Adj}(\bar{\mathcal{G}}, Y)$ *or* $\text{Adj}(\bar{\mathcal{G}}, X) \cap \text{Adj}(\bar{\mathcal{G}}, Z) \cap \text{Adj}(\bar{\mathcal{G}}, Y) \neq \emptyset$
7        Check the edge between $X$ and $Z$
8 Orient the new edges using the orientation rules in (Spirtes et al., 2000).   `// Orientation`

---

**Algorithm 5:** The Iterative PC Algorithm

---

**Input:** A set of variables $\mathcal{X}$ and their joint probability distribution $P$
**Output:** A partially directed acyclic graph

1 $\text{Sepset} = \emptyset$
2 $\hat{\mathcal{X}} = \{X_1, X_2\}$
3 $\mathcal{G}, \text{Sepset} \leftarrow PC(\hat{\mathcal{X}})$
4 **while** $\mathcal{X} \setminus \hat{\mathcal{X}} \neq \emptyset$ **do**
5    $X \in \mathcal{X} \setminus \hat{\mathcal{X}}$
6    $\bar{\mathcal{G}}, \text{Sepset} \leftarrow ExtendablePC(\mathcal{G}, X, \text{Sepset})$
7    $\hat{\mathcal{X}} \leftarrow \hat{\mathcal{X}} \cup \{X\}$
8    $\mathcal{G} \leftarrow \bar{\mathcal{G}}$
9 Orient the edges using the orientation rules in (Spirtes et al., 2000).    `// Orientation`

---

**Algorithm 6:** The PC Algorithm

---

**Input:** A set of variables $\mathcal{X}$ and their joint probability distribution $P$
**Output:** A partially directed acyclic graph

1 Form the complete undirected graph $\mathcal{G}$ over nodes $\mathcal{X}$;
2 $\text{Sepset}(X, Y) = \emptyset$ for all $X, Y \in \mathcal{X}$;
3 $m = 0$
4 **while** *maximum node degree in $\mathcal{G}$ is greater than $m$* **do**
5    **for** $X \in \mathcal{X}$          `// CI tests`
6      **for** $Y \in \text{Adj}(\mathcal{G}, X)$
7        **for** $\mathcal{U} \subseteq \text{Adj}(\mathcal{G}, X) \setminus \{Y\}$ *and* $|\mathcal{U}| = m$
8          **if** $X \perp Y \mid \mathcal{U}$
9            Remove the edge $X - Y$ from $\mathcal{G}$;
10            $\text{Sepset}(X, Y) \leftarrow \mathcal{U}$;
11    $m = m + 1$;
12 Orient the edges using the orientation rules in (Spirtes et al., 2000).    `// Orientation`

---

---

**Algorithm 7:** The Extendable exhaustive search structure learning algorithm

---

**Input:** A new variable $Y$ and a structure $\hat{\mathcal{G}}$ over $\mathcal{X}$
**Output:** A P-map $\bar{\mathcal{G}}$ over $\bar{\mathcal{X}} = \mathcal{X} \cup \{Y\}$

1   Form $\bar{\mathcal{G}}$ as the union of $\hat{\mathcal{G}}$ and one-point graph $Y$
2   $i = 1$
3   **while** $i < I$ **do**
4     $\mathcal{N}_{\bar{\mathcal{G}}} \leftarrow DAG\ Finder(\bar{\mathcal{G}})$
5     $\mathcal{N}_{\bar{\mathcal{G}}} \leftarrow \mathrm{T}(\hat{\mathcal{G}}, Y, \mathcal{N}_{\bar{\mathcal{G}}})$                   `// By T-function in algorithm 8`
6     $\bar{\mathcal{G}} \leftarrow \arg\max_{\mathcal{G} \in \mathcal{N}_{\bar{\mathcal{G}}}} \mathrm{Score}_{\mathrm{BIC}}(\mathcal{G})$
7     $i \leftarrow i + 1$

---

**Algorithm 8:** T-function (search space trimming)

---

**Input:** $Y$, graph structure $\hat{\mathcal{G}}$ over $\mathcal{X}$, and set of initial DAGs over $\bar{\mathcal{X}}$ denoted as $\mathcal{S}_{\bar{\mathcal{X}}}$
**Output:** $\mathcal{S}_{\bar{\mathcal{X}}}$ for $\bar{\mathcal{X}} = \mathcal{X} \cup \{Y\}$

1   **for** $\bar{\mathcal{G}} \in \mathcal{S}_{\bar{\mathcal{X}}}$
2     **for** $X_i, X_j \in \mathcal{X}$
3       **if** $X_i \notin \mathrm{Adj}(\hat{\mathcal{G}}, X_j)$ *and* $X_i \in \mathrm{Adj}(\bar{\mathcal{G}}, X_j)$
4         Delete $\bar{\mathcal{G}}$ from $\mathcal{S}_{\bar{\mathcal{X}}}$
5       **if** $(X_i \rightarrow X_j \leftarrow X_k) \in \hat{\mathcal{G}}$ *and* $((X_i \leftarrow X_j \leftarrow X_k) \in \bar{\mathcal{G}}$ *or* $(X_i \leftarrow X_j \rightarrow X_k) \in \bar{\mathcal{G}})$
6         Delete $\bar{\mathcal{G}}$ from $\mathcal{S}_{\bar{\mathcal{X}}}$
7       **if** $X_i \in \mathrm{Adj}(\hat{\mathcal{G}}, X_j)$ *and* $X_i \notin \mathrm{Adj}(\bar{\mathcal{G}}, X_j)$
8         **if** *Edges between* $X_i, X_j$ *and* $Y$ *do not form a structure similar to any of the structures in Fig. 1*
9           Delete $\bar{\mathcal{G}}$ from $\mathcal{S}_{\bar{\mathcal{X}}}$
10      **if** $X_i \notin \mathrm{Adj}(\hat{\mathcal{G}}, X_j)$ *and* $X_i, X_j \in \mathrm{Adj}(\bar{\mathcal{G}}, Y)$ *and* $(X_i \rightarrow Y \leftarrow X_j) \notin \bar{\mathcal{G}}$
11        Delete $\bar{\mathcal{G}}$ from $\mathcal{S}_{\bar{\mathcal{X}}}$
12      **if** $X_i \in \mathrm{Adj}(\hat{\mathcal{G}}, X_j)$ *and* $X_i, X_j$ *are collider nodes in* $\bar{\mathcal{G}}$ *and* $(X_i \leftarrow Y \rightarrow X_j) \notin \bar{\mathcal{G}}$
13        Delete $\bar{\mathcal{G}}$ from $\mathcal{S}_{\bar{\mathcal{X}}}$
14   Return $\mathcal{S}_{\bar{\mathcal{X}}}$

---

