# OpenReview forum: "Extendable and Iterative Structure Learning Strategy for Bayesian Networks"
_ICLR.cc/2025/Conference — ICLR 2025 Poster_

### Official Review · Reviewer_dGLi · 2024-11-01

**Soundness:** 3
**Presentation:** 3
**Contribution:** 3
**Rating:** 8
**Confidence:** 3

**Summary:**

This paper presents an efficient method for updating Bayesian network structures as new variables are introduced, eliminating the need for retraining from scratch. The approach reduces computational costs by up to 1300x without sacrificing accuracy. The authors also propose an iterative strategy that builds the network iteratively, thereby offering runtime benefits comparable to common algorithms like PC while maintaining accuracy. This scalable approach is well-suited for real-time and high-dimensional applications.

**Strengths:**

1) A simple iterative type algorithm for learning bayesian networks, which is computationally efficient.
2) Insights on unidentifiability and its relationship to faithfulness of the graph.
3) Backed up by theoretical claims

**Weaknesses:**

n/a

**Questions:**

n/a

---

### Official Review · Reviewer_hH34 · 2024-11-03

**Soundness:** 2
**Presentation:** 2
**Contribution:** 2
**Rating:** 3
**Confidence:** 2

**Summary:**

The paper proposes an extendable stracture learning method for Bayesian networks, which updates an existing network by adding new variables. The authors also propose a iterative approach for structured learning by starting with a small set and adding the remaining variables to the P-map graph. The authors run the extendable PC algorithm on multiple datasets, and show that their approach requires fewer number of CI tests compared to the original approaches.

**Strengths:**

- The paper is well organized. The notations and definitions are mostly self contained.
- The authors provide experiments on multiple datasets, and show that their approach save significant runtime because of the fewer number of CI tests required in the proposed iterative method.
- The proposed method seems straightforward and easy to implement in practice.

**Weaknesses:**

I am not an expert on this line of literature. My main concern is that the paper seems very heuristic with no formal guarantees:
- While the experimental results look good, I wonder if the proposed method of extendable PC has any consistency, faithfulness, or optimality guarantee.
- The results in Table 2 - 3 suggest that extendable PC always has a better runtime with fewer number of CI tests compared to PC. Can that be proved?
- The result in Table 5 shows that the proposed iterative PC does not always require fewer CI tests compared to PC. Under what statistical or topological conditions will that happen? In my opinion this is be the bigger risk, because without any theoretic characterization, this shows the possibility that the proposed approach does not generalize.

**Questions:**

- Is there any bound on the total number of CI tests required for running Algorithm 3?

---

> ### Author Response · Authors · 2024-11-21
>
> **W1. While the experimental results look good, I wonder if the proposed method of extendable PC has any consistency, faithfulness, or optimality guarantee.**
>
> In the new version, we proved that the output of the extendable algorithm is a P-map; see the following Theorem.
>
> Theorem. The output of Algorithms 1 and 5 is a P-map.
>
> Proof. The proof is a straightforward conclusion using Lemma 1, Lemma 3, and Lemma 4 (in Appendix). Lemma 4 shows that adding a new variable cannot add an edge between two nodes. Hence, according to Lemma 1, the output skeleton of the proposed extendable algorithm finds the skeleton of the true DAG. Then, Lemma 3 shows that all collider nodes were found correctly by the proposed algorithm. So the output PDAG for constraint-based algorithms such as Algorithm 1 is a P-map structure. $\square$
>
> The proof is a straight conclusion using Lemma 1, Lemma 3, and Lemma 4 (in Appendix). Lemma 4 leads to the fact that adding a new variable cannot add an edge between two nodes. According to Lemma 1, and using the fact, the output skeleton of the proposed extendable algorithm finds the skeleton of true DAG. Then, Lemma 3 shows that all collider nodes were found correctly by the proposed algorithm. So the output PDAG for constraint-based algorithms such as Algorithm 1 is a P-map structure.

---

> ### Author Response · Authors · 2024-11-21
>
> **W2. The results in Table 2 - 3 suggest that extendable PC always has a better runtime with fewer number of CI tests compared to PC. Can that be proved?**
>
> In section 3.1 of the new version, lines 246-269 (or old version lines 233-256), we discussed the number of CI tests for the extendable PC compared to the PC algorithm and showed that the number of CI tests for the proposed algorithm is always fewer than the PC and summarized the results in Table 1. The revised version includes Proposition 2  to explain this fact clearly.

---

> ### Author Response · Authors · 2024-11-21
>
> **W3. The result in Table 5 shows that the proposed iterative PC does not always require fewer CI tests compared to PC. Under what statistical or topological conditions will that happen? In my opinion this is be the bigger risk, because without any theoretic characterization, this shows the possibility that the proposed approach does not generalize.**
>
> Using an iterative approach, at each step, we leverage information about the relationships between nodes from the previous graph to determine the current graph. Since the number of nodes impacts the number of conditional independence (CI) tests, fewer nodes result in fewer CI tests when applying Lemma 1, which restricts the space of possible graphs. Also, the performance of the iterative algorithms depends on the order of selecting variables. According to Lemma 1 and Figure 1 (a-e), if the ordering is close to a topological causal ordering the performance of the iterative will be better. For example, consider a naive Bayes structure with $n$ children $\{X_1,\cdots, X_n\} $ and a parent node $Y$. With ordering like $\langle X_1,\cdots, X_n, Y\rangle  $, before dealing with $Y$, the iterative algorithm will first produce the complete graph over $\{X_1,\cdots, X_n\} $. So this ordering will require an exponential number of CI tests. However, with this ordering $\langle Y, X_1,\cdots, X_n\rangle $ the number of CI tests for Algorithm 3 when using Algorithm 1 as the extendable P-map learner will be fewer than the previous ordering.
>
> The new version includes Theorem 2 which proves that there is an order of variables for which the required number of CI tests for the iterative PC algorithm is fewer than for the PC algorithm. The main idea of the proof is based on Lemma 1. By using a topological causal ordering on the variables, according to Lemma 1, the variable $Y$ in Figure 1 (a-e) is chosen sooner than the children variables. This means the extra spurious edges will not occur in the graph during the entire process. In addition,  we proved that the number of CI tests for extendable PC is fewer than the PC to check each edge. Thus the number of CI tests for the iterative PC will be fewer than for the PC.

---

> ### Author Response · Authors · 2024-11-21
>
> **Q1. Is there any bound on the total number of CI tests required for running Algorithm 3?**
>
> The number of CI tests for the iterative PC algorithm depends on the order of variables. So according to the above discussion, we can find an order that guarantees reducing the number of CI tests in comparison with the PC.

---

### Official Review · Reviewer_wmKg · 2024-11-03

**Soundness:** 3
**Presentation:** 3
**Contribution:** 3
**Rating:** 6
**Confidence:** 3

**Summary:**

The paper studies the structure learning problem in Bayesian networks, i.e.. learning a directed acyclic graph (DAG) which defines the conditional probability distribution  over the given variables. Particularly, an extendable structure learning strategy is proposed to update an existing Bayesian network graph efficiently, when  a new variable is introduced. Two approaches (constraint-based and score-based) are discussed for extendable structure learning, which leverage the previous graph structure, and have   much lower computational cost compared to relearning the graph. It is then shown that these procedures can be used to design an iterative algorithm for structure learning. Numerical results on many graph datasets illustrate the performance of  the proposed method, and shows significant speedup over the approach where the graph is learned from scratch.

**Strengths:**

Strengths:
1.  Extendable structure learning for Bayesian networks is studied and two new approaches are proposed.
2. The new approaches achieve much lower runtime than relearning the graphs without prior structure.
3. Numerical results are presented on  many graph datasets showing  significant speedup.

**Weaknesses:**

Weakness:
1. Performance guarantees for the proposed method are not presented.
2. Details about a key assumption can be further discussed.

**Questions:**

I have the following comments about the paper:

1. Performance guarantees such as error analysis, or on how to check whether the algorithm has converged to the correct graph structure are not discussed.
A discussion on how to verify the correctness of  the P-map finder algorithm used should be added. If the iterative approach is used, how does the error accumulate?
Are the conditions in Lemma 1 and 2 sufficient for  the P-map finder algorithm output the true structure?


2. The key assumption that when a new variable $Y$ is added to the existing set $X$, that no new edges get assigned between the elements of $X$, this assumption needs further explanation. This seems to be a necessary condition of the algorithm to have low computational cost. In many situations, the introduction of a new variable might introduce new dependencies between existing nodes, e.g., in root-cause analysis, causal learning, molecular prediction, and others. Also, such situations could occur in time evolving DAGs. Further discussion on these will illustrate the applicability of the proposed method to different problems.


3. Minor Comment:
i. In abstract, algorithm named PC might not be known to general readers. Similarly, FCI. Some of these acronyms are not defined.

---

> ### Author Response · Authors · 2024-11-21
>
> **Q1. Performance guarantees such as error analysis, or on how to check whether the algorithm has converged to the correct graph structure are not discussed. A discussion on how to verify the correctness of the P-map finder algorithm used should be added. If the iterative approach is used, how does the error accumulate? Are the conditions in Lemma 1 and 2 sufficient for the P-map finder algorithm output the true structure?**
>
> In the revised version, we proved a theorem that shows the output of the extendable algorithm is a P-map.
> The proof is a straightforward conclusion using Lemma 1, Lemma 3, and Lemma 4 (in Appendix). Lemma 4 shows that adding a new variable cannot add an edge between two nodes. Hence, according to Lemma 1, the output skeleton of the proposed extendable algorithm finds the skeleton of the true DAG. Then, Lemma 3 shows that all collider nodes were found correctly by the proposed algorithm. So the output PDAG for constraint-based algorithms such as Algorithm 1 is a P-map structure.
>
> Theorem. The output of Algorithms 1 and 5 is a P-map.
>
> Proof. The proof is a straightforward conclusion using Lemma 1, Lemma 3, and Lemma 4 (in Appendix). Lemma 4 shows that adding a new variable cannot add an edge between two nodes. Hence, according to Lemma 1, the output skeleton of the proposed extendable algorithm finds the skeleton of the true DAG. Then, Lemma 3 shows that all collider nodes were found correctly by the proposed algorithm. So the output PDAG for constraint-based algorithms such as Algorithm 1 is a P-map structure. $\square$
>
> Error accumulation is a critical challenge in constraint-based approaches and non-exhaustive score-based searches, such as greedy search. These methods rely on iterative updates where decisions to add or delete edges are based on the previous structure. Errors in earlier iterations propagate and compound, affecting subsequent steps. Similarly, iterative algorithms inherit this issue as they rely on standard approaches to refine structures, further perpetuating inaccuracies introduced in prior iterations.

---

> ### Author Response · Authors · 2024-11-21
>
> **Q2. The key assumption that when a new variable $Y$ is added to the existing set $X$, that no new edges get assigned between the elements of $X$, this assumption needs further explanation. This seems to be a necessary condition of the algorithm to have low computational cost. In many situations, the introduction of a new variable might introduce new dependencies between existing nodes, e.g., in root-cause analysis, causal learning, molecular prediction, and others. Also, such situations could occur in time evolving DAGs. Further discussion on these will illustrate the applicability of the proposed method to different problems.**
>
> Thank you for your comments.  Note, however, that this claim about “adding a new variable cannot add any new edges to the prior model”, is not an assumption, but is a provable proposition, for Bayesian networks. We have now added the following proposition.
>
> Proposition. Consider $\bar G$ is a P-map over $\bar X$ and $G$ is a graph over $X$. If $X_j$ is an adjacent of $X_i$ in $\bar G$, then $X_j$ is an adjacent of $X_i$ in $G$.
>
> Proof.
>     If $X_j$ and $X_i$ are not adjacent in $G$, according to Lemma 4, there is a subset $U \subseteq X\setminus\{X_i,X_j\}$ such that $X_i\perp X_j | U$. Because $\bar G$ is a P-map and due to faithfulness assumption $X_i$ and $X_j$ are d-separated by $U$ in $\bar G$. Then there is no edge between $X_i$ and $X_j$ in $\bar G$.
> $\square$
>
> One of the basic theorems in Bayesian network structure learning is Lemma 4 (in appendix), which uses the Markov condition and faithfulness assumption. According to the lemma 4, if there is a subset of variables $\mathcal{U}$ such that two variables $X$ and $Y$ would be independent given it, so there is no edge between $X$ and $Y$. So we have only two basic assumptions Markovness and fatefulness which hold for almost all Bayesian networks except a zero-measure set of probability distributions. On the other side, the claim "adding a new variable cannot add a new edge or dependency" is a direct result of Lemma 4. So, we only used the Markov condition and faithfulness assumption for this claim. And this is not an extra assumption.
>
> We added a discussion about this issue. However, we are interested in knowing more about many situations in which the introduction of a new variable might introduce new dependencies between existing nodes, e.g., in root-cause analysis, causal learning, and molecular prediction. Maybe these problems cannot be modeled by Bayesian networks or the faithfulness assumption is violated. Of course, in time-evolving DAGs, since the structure can change over time, the proposed extendable approach cannot guarantee the output is a P-map for all times.

---

> ### Author Response · Authors · 2024-11-21
>
> **Q3. Minor Comment: i. In abstract, algorithm named PC might not be known to general readers. Similarly, FCI. Some of these acronyms are not defined.**
>
> Thanks for noting these limitations.  The revised version expands these acronyms and also provides a high-level summary of these algorithms.

---

### Official Review · Reviewer_g7c8 · 2024-11-05

**Soundness:** 3
**Presentation:** 2
**Contribution:** 2
**Rating:** 6
**Confidence:** 3

**Summary:**

This paper provides a novel perspective to the Bayesian structure learning problem with an efficient mechanism to add new variables to an underlying graphical structure. Specifically, the learning strategy hinges on efficiently incorporating a new variable $Y$ into an existing Bayesian network ${\cal G}$ over the set of variables ${\cal X}$, which results in an. updated Bayesian network $\bar {\cal G}$ on the augmented set of variables ${\cal X} \cup Y$. This learning strategy is further extended to provide a novel learning paradigm for structure learning for Bayesian networks. Experiments demonstrate significant computational efficiency over existing state-of-the-art algorithms.

**Strengths:**

The problem studied in the paper is well-motivated and will be of interest to ML community. Recycling the existing information when new variables are added or revealed provides a novel perspective to the structure learning problem. The iterative structure learning algorithm is a valuable contribution. Experiments convincingly demonstrate the computational efficiency of the proposed approach.

**Weaknesses:**

I found the discussions and organization of Section 3 to be more convoluted than necessary. In particular, it would be helpful to have the relevance of Algorithms 2-4 explictly elucidated.

**Questions:**

1. Is the Extendable P-map learner in Algorithm 3 implemented using Algorithm 1?
2. What is Sepset(X,Y)?
3. An intuitive explanation for why the iterative learning algorithm outperforms PC algorithm while learning the complete graph is recommended.
4. How is this work positioned relative to the prior works that study the information-theoretic limits of Bayesian structure learning?


____________________________________________
POST AUTHOR REBUTTAL
I thank the authors for their detailed response. I will maintain my rating.

---

> ### Author Response · Authors · 2024-11-21
>
> **W1. I found the discussions and organization of Section 3 to be more convoluted than necessary. In particular, it would be helpful to have the relevance of Algorithms 2-4 explictly elucidated.**
>
> Thank you for your excellent suggestion. The revised version now better explains the extendable score-based approach.  Section 3.2 now states:
>
> Algorithm 2 represents a general extendable score-based algorithm that includes:
> (1) a search space trimming function (T-function in Algorithm 4) that restricts the graph search space, based on the analysis from Lemmas 1 - 3; and (2) a score-based P-map finder (for example global minimization of the BIC score), that finds the best graph within the restricted search space.

---

> ### Author Response · Authors · 2024-11-21
>
> **Q1. Is the Extendable P-map learner in Algorithm 3 implemented using Algorithm 1?**
>
> Yes, Algorithm 1 is an extendable constraint-based algorithm based on the PC algorithm. So Algorithm 3 can use Algorithm 1 as an extendable P-map learner or use Algorithm 7, the extendable Hill-climbing that is an extendable score-based algorithm.

---

> ### Author Response · Authors · 2024-11-21
>
> **What is Sepset(X,Y)?**
>
> In the revised version, (discussing Algorithm 1) explicitly introduces $Sepset(X,Y)$. In general, for all $X,Z \in \mathcal{X}$, $Sepset(X,Z)$ is a subset of variables $\mathcal{X}\setminus \{ X,Z \}$ that leads $X\perp Z | Sepset (X,Z)$. In other words, $Sepset(X,Z)$ is a separating set for $X$ and $Z$.

---

> ### Author Response · Authors · 2024-11-21
>
> **An intuitive explanation for why the iterative learning algorithm outperforms PC algorithm while learning the complete graph is recommended.**
>
> The revised version has been modified to explain this issue. Section 3.3 states that:
> Using an iterative approach, at each step, we leverage information about the relationships between nodes from the previous graph to determine the current graph. Since the number of nodes impacts the number of conditional independence (CI) tests, fewer nodes result in fewer CI tests when applying Lemma 1, which restricts the space of possible graphs. Also, the performance of the iterative algorithms depends on the order of selecting variables. The performance of the iterative algorithms depends on the order of selecting variables. According to Lemma 1 and Figure 1 (a-e), if the ordering is close to causal ordering the performance of the iterative will be better. For example, consider a naive Bayes structure with $n$ children $\{X_1,\cdots, X_n\} $ and a parent node $Y$. With ordering like $\langle X_1,\cdots, X_n, Y\rangle  $, before dealing with $Y$, the iterative algorithm will first produce the complete graph over $\{X_1,\cdots, X_n\} $. So this ordering will require an exponential number of CI tests. However, with this ordering $\langle Y, X_1,\cdots, X_n\rangle $ the number of CI tests for Algorithm 3 when using Algorithm 1 as the extendable P-map learner will be fewer than the previous ordering.

---

> ### Author Response · Authors · 2024-11-21
>
> **How is this work positioned relative to the prior works that study the information-theoretic limits of Bayesian structure learning?**
>
> One can use the information-theoretic limits to find a bound on the minimum number of new instances needed to recover the true DAG once a new variable is added to the previous set of variables. For example, for the case with discrete variables and where the conditional probability distributions are modeled with conditional probability tables (CPT), the addition of one variable to the existing $N$ variables increases the bound for a non-sparse DAG by $\dfrac{2N+1}{\theta_{min}}$  instances, where $\theta_{min}$ is the minimum probability value in the conditional probability tables [1].
>
> [1] Information-theoretic limits of Bayesian network structure learning

---

### Official Review · Reviewer_RR2Y · 2024-11-12

**Soundness:** 3
**Presentation:** 3
**Contribution:** 3
**Rating:** 5
**Confidence:** 3

**Summary:**

This paper proposes an iterative/incremental algorithm to learn the structure of a Bayeisan network from observed data.
The algorithm added one variable at a time, and modify the previously learn structure accordingly.
The novelty is to prove that adding one variable only leads to deletion of previous edges, and therefore trim the search space of possible networks.

In particular, Lemma 1 shows that if the new variable Y satisfies some properties, a certain kind of edges in the old graph can be safely deleted. The paper introduced constrained-based and score-based approaches to utilize this lemma. The results is a reduction of number of CI tests.

Experiments on 13 datasets shows that the running time is significantly reduced without compromising accuracy of the learned structure.

**Strengths:**

+ structure learning is a challenging problem due to exponentially large search space, and the paper makes progress in speeding up the search without sacrificing accuracy.
+ the proof is solid and the presentation is clear.
+ strong experimental results in accuracy and running time.

**Weaknesses:**

- The scope of the paper (symbolic Baysian network) may not fit ICLR conference (representation learning).
- Insufficient discussion of related work. There shall be a large number of related work on incremental structure learning, while the submission only cite two most relevant ones ((Kocacoban & Cussens, 2019) and (Alcobe, 2005)). This makes the contribution of the submission less clear.

**Questions:**

- How the score-based and constraint-based methods are used in the extendable PC algorithm?
- Are the more existing work that incrementally learn the structure? What are their strengths and weaknesses?

---

> ### Author Response · Authors · 2024-11-21
>
> **W1. The scope of the paper (symbolic Baysian network) may not fit ICLR conference (representation learning).**
>
> Please note that ICLR has published many  Bayesian networks, causal discovery, structure learning, and Bayesian network inference, including the following, in years 2023 and 2024. This demonstrates that our proposed paper about structure learning Bayesian networks aligns well with the scope of the ICLR conference.
>
> * Tractable Probabilistic Graph Representation Learning with Graph-Induced Sum-Product Networks
> * Federated Causal Discovery from Heterogeneous Data
> * Learning Large DAGs is Harder than you Think: Many Losses are Minimal for the Wrong DAG
> * Constraint-Free Structure Learning with Smooth Acyclic Orientations
> * Structural Estimation of Partially Observed Linear Non-Gaussian Acyclic Model: A Practical Approach with Identifiability
> * Causal Structure Recovery with Latent Variables under Milder Distributional and Graphical Assumptions
> * Diffusion Models for Causal Discovery via Topological Ordering
> * BayesDiff: Estimating Pixel-wise Uncertainty in Diffusion via Bayesian Inference
> * Unified Generative Modeling of 3D Molecules with Bayesian Flow Networks
> * Training Bayesian Neural Networks with Sparse Subspace Variational Inference

---

> > ### Comment · Reviewer_RR2Y · 2024-11-28
> > **Related work on ICLR well-received**
> >
> > No further problem with this question.

---

> ### Author Response · Authors · 2024-11-21
>
> **W2. Insufficient discussion of related work. There shall be a large number of related work on incremental structure learning, while the submission only cite two most relevant ones ((Kocacoban & Cussens, 2019) and (Alcobe, 2005)). This makes the contribution of the submission less clear.**
>
> Thank you for noting this omission.  The revised paper now explicitly notes that all incremental/online methods for learning Bayesian network structures consider ways to incorporate a set of new instances to the current dataset – so going from $n$ instances (each with $k$ variables) to $n+1$ instances (over $k$ variables). By contrast, our proposed algorithm adds one (or more) new variables to the current set of variables – but over the same set of instances – so going from $n$ instances over $k$ variables, to $n$ instances over $k+1$ variables. These are two different problems, requiring a new strategy in Bayesian network structure learning.
>
> Imagine we have a dataset $S[X]$ of 100 instances, each describing a different person, by providing the values for 10 features $X$ – 9 covariates describing a person – eg, age, sex, smoker, … – , as well as a 10th, for whether this person has a specific disease (say lung cancer).  Now let $BN( S[X] )$ be the  Bayesian Network (over 10 variables $X$), learned from those 100  instances.  Now imagine we decided to acquire other information about each person – say the expression of $Gene22$.  This means we can describe each patient with 11 features, $X’ = X \cup { Gene22 }$ – producing the 100-instance (11-d) dataset $S’[X’]$.  The challenge now is to produce a new Bayesian over the 11 features – call it $BN( S’[X’] )$, using this new $S’[X’]$ dataset, along with the earlier $BN( S[X] )$.
>
> Notice this task is fundamentally different from the more traditional online Bayesian network structure-learning task, which might go from this 100-instance dataset $S[X]$, by adding in one (or more) next 10-d instances.

---

> ### Author Response · Authors · 2024-11-21
>
> **Q1. How the score-based and constraint-based methods are used in the extendable PC algorithm?**
>
> The proposed extendable approach can be applied to all constraint-based and score-based algorithms that are guaranteed their output is a P-map. We developed “extendable PC” as an example of extendable constraint-based algorithms. The extendable score-based algorithm has been illustrated in Algorithm 2 in its general form. In addition, Algorithm 7 represents the extendable Hill-climbing as a special case of Algorithm 2.

---

> ### Author Response · Authors · 2024-11-21
>
> **Q2. Are the more existing work that incrementally learn the structure? What are their strengths and weaknesses?**
>
> To the best of our knowledge, there is no structure-learning method that can deal with the addition of a new variable (not instance) as explained in our comment above.

---

> > ### Comment · Reviewer_RR2Y · 2024-11-28
> > **More related work can be discussed and compared**
> >
> > Hi authors, thanks for your responses. The following work may be relevant to your work:
> >
> > Adaptive Exact Inference in Graphical Models. JMLR, 2011.
> > This paper focuses on inference (not structure learning). The beliefs are evaluated on a dynamic graph incrementally, and the speed-up can benefit structure learning. I am wondering if your method can benefit from such related work. Also, see page 3 of the paper where more incremental update algorithms are discussed.
> >
> > Furthermore, if the algorithm is constrained to learn tree structures only, how would the proposed algorithm be modified or upgraded to exploit such a constraint?

---

> > > ### Author Response · Authors · 2024-11-30
> > >
> > > Hi, thank you for your suggestion. As noted, the referenced paper develops an incremental inference method limited to tree structures. From our understanding, it focuses on updating the model when a dependency is added or removed, without introducing new variables. Thus, it is not directly related to our method, which specifically addresses the impact of adding new variables to the existing structure. However, your suggestion inspired us to extend our proposed extendable approach for inference in future works.
> > >
> > > Our proposed Lemmas 1-4 analyze the effect of adding new variables in a general form, making the algorithm applicable to tree structures as well. However, we agree that leveraging the constraints and specific properties of tree structures could further optimize performance for this case.

---

### Meta-Review · Area_Chair_DqwE · 2024-12-21

**Metareview:**

This paper proposes a method for ad-hoc addition of new nodes in bayesian structure learning, which is quite different from previous works. The reviewers are largely in agreement of the importance of the work, and the novelty of the proposed method. There were some minor concerns about the approach being too heuristic but the authors have addressed those.

**Additional Comments On Reviewer Discussion:**

The reviewers had concerns over the conference fit (bayesian learning vs representation learning), extension to another method (which is not in the scope of this paper IMO), the method being too heuristic and lacking theoretical properties, and some clarity issues. The authors have addressed the concerns well. I am ignoring the overly negative review (rating 3) by reviewer hH34 who agreed that the empirical results are good but wanted more theoretical contributions. The authors responded by adding some more theory during the rebuttal but the reviewer did not respond back.

---

### Decision · Program_Chairs · 2025-01-22

Accept (Poster)